# Investigation of Surface Quality for Minor Scale Diameter of Biodegradable Magnesium Alloys during the Turning Process Using a Different Tool Nose Radius

**Sophal Hai [1], Hwa-Chul Jung [2], Won-Hyun Shim [2] and Hyung-Gon Shin [1,*]**

[1] Department of Mechanical System Engineering, Jeonbuk National University, 567 Baekje-daero, Deokjin-gu, Jeonju-si 54896, Jeollabuk-do, Korea; khaisophal@naver.com
[2] U&I Corporation, Sandan-ro 76beon-gil(Rd), Uijeongbu-si 11781, Gyeonggi-do, Korea; kspkorea8787@gmail.com (H.-C.J.); wonhyun-shin@naver.com (W.-H.S.)
[*] Correspondence: vimission@jbnu.ac.kr; Tel.: +82-10-4654-8963

**Abstract:** The main objective of the study is to analyze the various cutting parameters to investigate the surface quality of the minor scale diameter of magnesium alloy in the dry turning process using a different tool nose radius ($r$). The surface roughness ($R_a$) was gauged, and micro-images produced by scanning electron microscopy (SEM) were reviewed to evaluate the machined surface topography. The analysis of variance (ANOVA), linear regression model and signal-to-noise (S/N) ratio were applied to investigate and optimize the experimental conditions for surface roughness. The study results imply that the feed rate and tool nose radius significantly affected the surface quality, but the spindle speed did not. The linear regression model is valid to forecast the surface roughness. The cutting parameters for optimum surface quality are a combination of a spindle speed of 710 rpm, a feed rate of 0.052 mm/rev and a tool nose radius of 1.2 mm. The machined surface topography contains the feed marks, micro-voids, material side and material debris, but they become smaller and decrease at a lower feed rate, larger tool nose radius and higher spindle speed. These results show the good surface quality of magnesium alloys in a dry turning process, which could be applied in implant, orthopedic and trauma surgery.

**Keywords:** biodegradable magnesium alloy; turning process; tool nose radius; surface quality; minor scale diameter

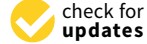

## 1. Introduction

Magnesium (Mg) alloys are broadly applied in various applications such as automotive, electronics, medicine, sports and household equipment. The advantages of this lightweight material include good castability and damping capacity, high specific strength and thermal conductivity, etc. [1–4]. In particular, the utilization of magnesium in medical applications (implant, orthopedic and trauma surgery) has significantly increased because it shows promise as a biodegradable metallic material for use in the human body [5–7]. The machined surface quality of magnesium plays a critical role in bone surgery (bone repair). The surface quality factor has the greatest effect on corrosion behavior as well as service life of the product [8,9]. Therefore, the machined surface quality of magnesium alloy needs investigation and analysis to improve its application performance.

During the last few decades, the advantages of magnesium alloy's mechanical properties have attracted many researchers. Due to the natural biocompatibility property of magnesium, it was used to replace titanium alloy and stainless steel for orthopedic applications such as bone substitutes, damaged bone tissue, joint arthroplasties, etc. [6,10,11]. Chakraborty Banerjee et al. [12] have reviewed the various types of implant biomaterials, which include magnesium alloys, titanium alloys, stainless steels and cobalt-chromium alloys. Magnesium alloys were concluded as the best promising candidate biomaterial for

temporary implants. Moreover, conversion coatings were recommended as the wide approach for biodegradable magnesium material in order to enhance its corrosion resistance during service in the human body. Song et al. [13] also reported that AZ91D magnesium alloys are potential implant biomaterials based on their advantageous biological properties (e.g., magnesium alloys have a biodegradable property in the fluid of the human body by corrosion, $Mg^{2+}$ is harmless to humans and the growth of new bone tissue can be accelerated).

For the orthopedic application, the machined surface quality of magnesium alloy is a strong concern because it has a close relationship to the service life and corrosion behavior of magnesium. In order to enhance the machined surface quality, many machined methods and parameters have been applied to evaluate and investigate the surface roughness. Bruschi et al. [14] have performed the turning operation in three environments such as dry, wet and under cryogenic cooling (liquid nitrogen). The AZ31 magnesium workpieces were formed in bars of a Ø30-mm diameter and 180-mm length. With the process using liquid nitrogen, the machined surface integrity was performed in the best condition and led to the improvement of the corrosion resistance of the machined surface. On the other hand, Patel et al. [15] observed the machined surface roughness based on tool nose radius and cutting parameters during a dry turning test on a AISI D2 steel workpiece. The surface roughness ($R_a$) was obtained at a lower value, 0.504 μm, since the nose radius increased. Dutta et al. [16] has performed the turning test on a magnesium workpiece (Ø36.5 mm in diameter and 200 mm in length) with a different tool nose radius. The study concluded that the tool nose radius was an important parameter to investigate and improve the machined surface quality. As the results show, the larger nose radius can produce a better surface roughness. Wojtowicz et al. [17] reported that surface integrity is a key factor in the machined component such as fatigue life. The tool nose radius performs as a cutting parameter that has significant effects on the machined surface roughness quality during the turning test of the magnesium material. The machined surface was better, since the tool nose radius increased. However, both researchers [16] and [17] revealed a lower surface roughness with the results range of 0.6–1.5 μm and 1.41–33.81 μm, respectively. Thus, the dry turning process using tool noses of varying radii was more challenging in order to enhance the surface roughness quality of magnesium. Moreover, for orthopedic and trauma surgery, the minor scale diameter of the magnesium bar is required for application in the human body for many cases.

The primary objective of this research is to investigate the machined surface quality of a biodegradable magnesium alloy during a dry turning process with the following various cutting parameters: tool nose radius ($r$), spindle speed ($N$) and feed rate ($f$). The minor scale diameters of the magnesium workpiece were machined by triangular uncoated carbide inserts with a different tool nose radius ($r$). The surface quality was investigated using a surface roughness tester (SJ-210, Mitutoyo, Busan, Korea) and scanning electron microscope (JSM-5900, JEOL, Tokyo, Japan). The linear regression model, ANOVA and S/N ratio were applied to analyze the surface roughness behavior.

## 2. Materials and Methods

### 2.1. Experiment Procedure

Figure 1 shows the schematic of the experimental setup that applies to the present research. The dry turning experiment was conducted on the NC lathe (HL460 × 1000GN, HWACHEON, Seoul, Korea) with a spindle power of 5.5 KW and maximum spindle speed of 1800 rpm. The 27 experimental samples were prepared for the turning test. After the turning process, the samples were removed from the NC lathe in order to investigate the machined surface quality. The machined surface quality was evaluated using a surface roughness tester and scanning electron microscope (SEM). The surface roughness tester (SJ-210) was employed to measure the machined surface roughness in different three zones ($R_{a1}$, $R_{a2}$ and $R_{a3}$) for one workpiece. The surface roughness ($R_a$) was calculated using $R_a = (R_{a1} + R_{a2} + R_{a3})/3$. The micro-images of the machined surface were evaluated using

SEM (JSM-5900) at 100× and 1000× magnifications. In this research, the experimental conditions are summarized in Table 1.

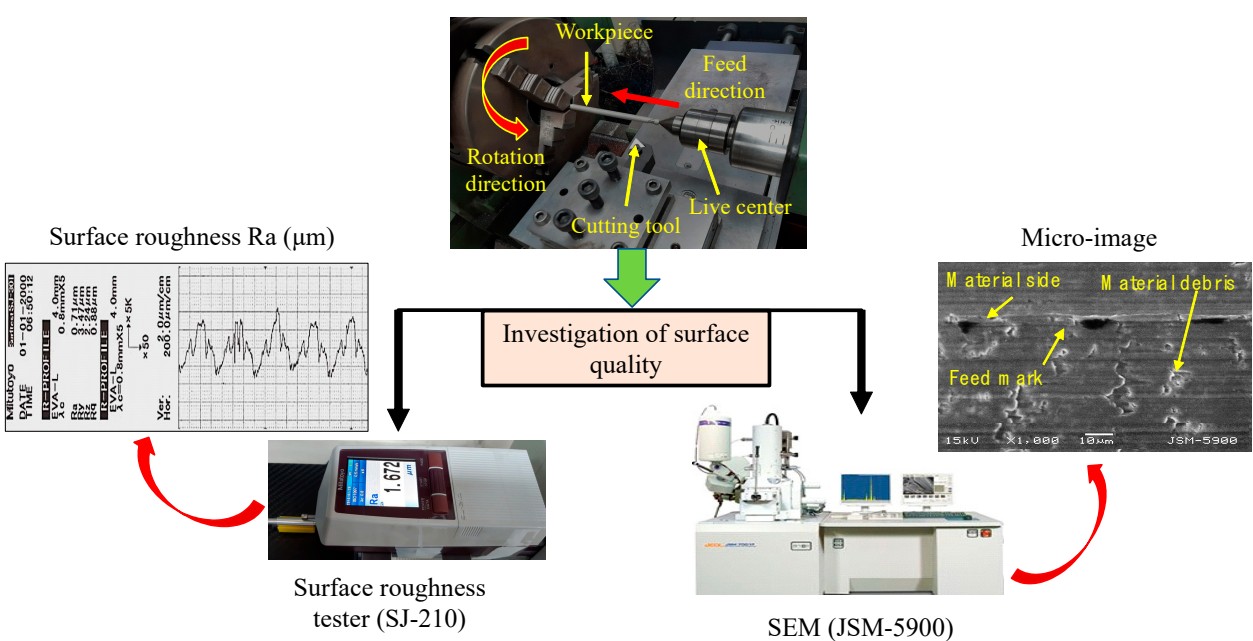

**Figure 1.** Schematic of experimental setup.

**Table 1.** Experimental conditions for dry turning process.

| Levels | Cutting Parameters | | | | Environment |
|---|---|---|---|---|---|
| | Tool Nose Radius $r$ (mm) | Feed Rate $f$ (mm/rev) | Spindle Speed $N$ (rpm) | Depth of Cut $d$ (mm) | |
| 1 | 0.4 | 0.052 | 510 | | |
| 2 | 0.8 | 0.105 | 710 | 0.25 | Dry turning |
| 3 | 1.2 | 0.209 | 1010 | | |

### 2.2. Materials and Cutting Tools

Magnesium alloys (Resomet) were used as the work material in this research, which was supplied by U&I corporation (Seoul, Korea). The chemical compositions were 4.72 wt.% Ca, 0.76 wt.% Zn and balanced wt.% Mg. The workpieces were formed with a Ø8-mm diameter and 120-mm length. The live center was used to support the workpiece for accurate machining on the axis and prevent the excessive deflection of the workpiece. The triangular uncoated carbide inserts of 11° clearance angle (Taegutec LTD., Daegu, Korea) were applied for the turning process. Three types of inserts $T_1$ (TPGN 160,304 K10), $T_2$ (TPGN 160,308 K10) and $T_3$ (TPGN 160,312 K10) were designed with different nose radii of $r_1 = 0.4$, $r_2 = 0.8$ and $r_3 = 1.2$ mm, respectively. The insert was changed for each experimental condition. The combination of the tool holder (CTGPR 2020 K16, Taegutec LTD., Daegu, Korea) and clamped insert resulted in −1° side cutting edge angle, 0° rake angle and 29° end cutting edge angle.

### 2.3. Statistical Analysis

The statistical analysis was utilized to investigate and optimize the experimental conditions for surface roughness. For this research, Minitab statistical software was used to perform the statistical analysis. An ANOVA was carried out to define which cutting parameters significantly affect the responses such as surface roughness, tool wear, tem-

perature, cutting force, etc. [18–20]. As shown in Section 3.2, degrees of freedom (DF) are used to calculate Adj MS; the adjusted sum of squares (Adj SS) measures the number of additional cutting parameter in the response, the adjusted mean squares (Adj MS) is the Adj SS divided by the DF, the F-Value is Adj MS divided by the Error value and the P-Value is used to indicate which cutting parameters significantly affect the responses. If the P-Value of a cutting parameter is less than or equal to 0.05, it indicates that the cutting parameter significantly affects a response [21,22].

The S/N ratio is a metric designed using Taguchi methodology to optimize the robustness of a product or process. The response values of surface roughness ($R_a$) were transformed in the S/N ratio. The S/N ratios are classified in the following three types: larger-is-better, smaller-is-better and nominal-is-best. The type of S/N ratio is selected to analyze the experiment data according to the objective of the study. For this research, the S/N ratio of the smaller-is-better type is carried out to analyze the cutting parameters and surface quality because the minimized value of surface roughness ($R_a$) is required. Hence, the S/N ratio of smaller-is-better type is described as the following equation [23,24]:

$$\text{Smaller} - \text{is} - \text{better type}: \text{S/N} = -10 \times \log\left(\frac{1}{n}\sum y^2\right) \tag{1}$$

where $y$ is the observed data and $n$ is the number of observations of the experiment.

## 3. Results and Discussion

### 3.1. Effects of Cutting Parameters (f, N, r) on Surface Roughness ($R_a$)

In this research, the dry turning process was performed in the L27 orthogonal array, and the results of surface roughness ($R_a$) for the 27 experiments are shown in Table 2. The values of machined surface roughness ($R_a$) are illustrated in the ranges of 0.46–2.9, 0.4–1.68 and 0.38–1.05 μm for various tool nose radii $r_1$, $r_2$ and $r_3$, respectively. Figure 2a–c shows the effect of spindle speed (N) on surface roughness ($R_a$) using the tool nose radii of $r_1$, $r_2$ and $r_3$, respectively, during the dry turning process. The general trend is that the values of surface roughness ($R_a$) slightly changed with the change of spindle speed (N). From the feed rate of 0.209 mm/rev, the surface roughness ($R_a$) increased with an increase in the spindle speed (N). These results have a similar trend compared with the study by Sahithi [25], which reported that the best surface finish was machined at the lowest spindle speed (500 rpm). From the feed rate of 0.105 mm/rev, the surface roughness ($R_a$) slightly decreased with the increase in the spindle speed range from 510 to 710 rpm and then increased at a spindle speed of 1010 rpm. However, from the lowest feed rate of 0.052 mm/rev, the surface roughness ($R_a$) gradually decreased with an increase in the spindle speed (N). Youselfi et al. [26] has reported that the rise of the spindle speed generated the linear decrease in surface roughness due to the cutting force reduced at a higher cutting speed. The thermal softening of the material occurred at a high cutting speed and then led to a decrease in cutting force. Thus, it results in a good surface roughness. However, in the case of using ceramic and coated carbide inserts, the surface roughness increased with an increase in the cutting speed due to the built-up edge formation (BUE) that occurred during the turning test. From the feed rate of 0.209 mm/rev, the surface roughness ($R_a$) is significantly higher compared to the surface roughness from the feed rates of 0.105 and 0.052 mm/rev. From the feed rates of 0.105 and 0.052 mm/rev, the surface roughnesses ($R_a$) are close for all the experimental conditions. Figure 2d–f shows the effect of feed rate (f) on surface roughness ($R_a$) using the tool nose radii of $r_1$, $r_2$ and $r_3$, respectively. For the general trend, the increase in the feed rate (f) generated a higher surface roughness ($R_a$) in all the experimental conditions. As shown in Figure 2, the feed rate (f) has a greater effect than spindle speed (N) on surface roughness ($R_a$). Khanna et al. [27] also observed that an increase in the feed rate caused the increment of surface roughness due to the feed marks effect. Bouacha et al. [28] reported that the feed rate strongly effects the surface roughness due to its primary function, known as the theoretical geometrical surface roughness. Dinesh et al. [29] found that the surface roughness increased with an

increase in the feed rate, owing to the rise of the material removal rate at a specific speed. The rise of material removal led to increased friction. The friction caused tool chatter and then produced a rough surface. The surface roughness ($R_a$) slightly increased from the feed rates of 0.052 to 0.105 mm/rev. However, the surface roughness ($R_a$) significantly increased from the feed rates of 0.105 to 0.209 mm/rev. This is due to the incremental rise in the feed rate from 0.105 to 0.209 mm/rev being higher than the feed rate from 0.052 to 0.105 mm/rev by about two times. Hence, it strongly modifies the feed mark and leads to deeper furrows.

**Table 2.** Experimental results of surface roughness ($R_a$).

| Cutting Tool | Run No. | Cutting Parameters | | | Surface Roughness $R_a$ (µm) |
| | | Spindle Speed $N$ (rpm) | Feed Rate $f$ (mm/rev) | Nose Radial $r$ (mm) | |
|---|---|---|---|---|---|
| T$_1$ | 1 | 510 | 0.052 | | 0.54 |
| | 2 | 510 | 0.105 | | 0.85 |
| | 3 | 510 | 0.209 | | 2.71 |
| | 4 | 710 | 0.052. | | 0.49 |
| | 5 | 710 | 0.105 | $r_1$ = 0.4 | 0.80 |
| | 6 | 710 | 0.209 | | 2.83 |
| | 7 | 1010 | 0.052 | | 0.46 |
| | 8 | 1010 | 0.105 | | 0.85 |
| | 9 | 1010 | 0.209 | | 2.90 |
| T$_2$ | 10 | 510 | 0.052 | | 0.43 |
| | 11 | 510 | 0.105 | | 0.54 |
| | 12 | 510 | 0.209 | | 1.75 |
| | 13 | 710 | 0.052 | | 0.41 |
| | 14 | 710 | 0.105 | $r_2$ = 0.8 | 0.50 |
| | 15 | 710 | 0.209 | | 1.66 |
| | 16 | 1010 | 0.052 | | 0.40 |
| | 17 | 1010 | 0.105 | | 0.52 |
| | 18 | 1010 | 0.209 | | 1.68 |
| T$_3$ | 19 | 510 | 0.052 | | 0.46 |
| | 20 | 510 | 0.105 | | 0.51 |
| | 21 | 510 | 0.209 | | 0.93 |
| | 22 | 710 | 0.052 | | 0.39 |
| | 23 | 710 | 0.105 | $r_3$ = 1.2 | 0.45 |
| | 24 | 710 | 0.209 | | 1.00 |
| | 25 | 1010 | 0.052 | | 0.38 |
| | 26 | 1010 | 0.105 | | 0.47 |
| | 27 | 1010 | 0.209 | | 1.05 |

Figure 3 shows the effect of the tool nose radius ($r$) on surface roughness ($R_a$) for different spindle speeds ($N$) and feed rates ($f$). Figure 3a–c shows the effect of the tool nose radius ($r$) on surface roughness ($R_a$) for different spindle speeds of 510, 710 and 1010 rpm, respectively. Generally, the surface roughness ($R_a$) decreased with the increase in the tool nose radius ($r$). This result can be explained by the equation that the tool nose radius ($r$) has a correlation with surface roughness ($R_a$): $R_a = f^2/32r$, where $f$ is the feed rate [30]. It clearly reveals that the surface roughness become better with a larger tool nose radius. Dutta et al. [16] declared that surface roughness was enhanced when using a larger tool nose radius in the turning process due to reducing the pressure. It results in decreased vibration, then leads to an improving surface roughness. Moreover, the larger tool nose radius can endure a stronger cutting force during the machining process, which results in a lower rate of flank wear and chipping. Thus, the larger tool nose radius can be applied to enhance the surface quality. From the feed rates of 0.105 and 0.209 mm/rev, the surface roughness ($R_a$) rapidly decreased with the increase in the tool nose radius ($r$). However, from the feed rate of 0.052 mm/rev, the surface roughness ($R_a$) gradually decreased when

the tool nose radius (*r*) became larger as a result of the incremental rise in feed rate from 0.105 to 0.209 mm/rev being higher than the feed rate from 0.052 to 0.105 mm/rev. In comparison with the tool nose radius (*r*) from 0.4 to 0.8 mm, the surface roughness ($R_a$) is more significantly decreased from 0.8 to 1.2 mm. Figure 3d–f shows the effect of tool nose radius (*r*) on surface roughness ($R_a$) for different feed rates of 0.052, 0.105 and 0.209 mm/rev, respectively. The surface roughness ($R_a$) decreased with the increase in the tool nose radius (*r*). Figure 3d illustrates that the surface roughness ($R_a$) gradually decreased with the increase in the tool nose radius. Figure 3e,f shows that the surface roughness ($R_a$) strongly decreased with an increase in the tool nose radius. Figure 3e,f demonstrate that the tool nose radius strongly effects the surface roughness in all the feed rates of 0.052, 0.105 and 0.209 mm/rev.

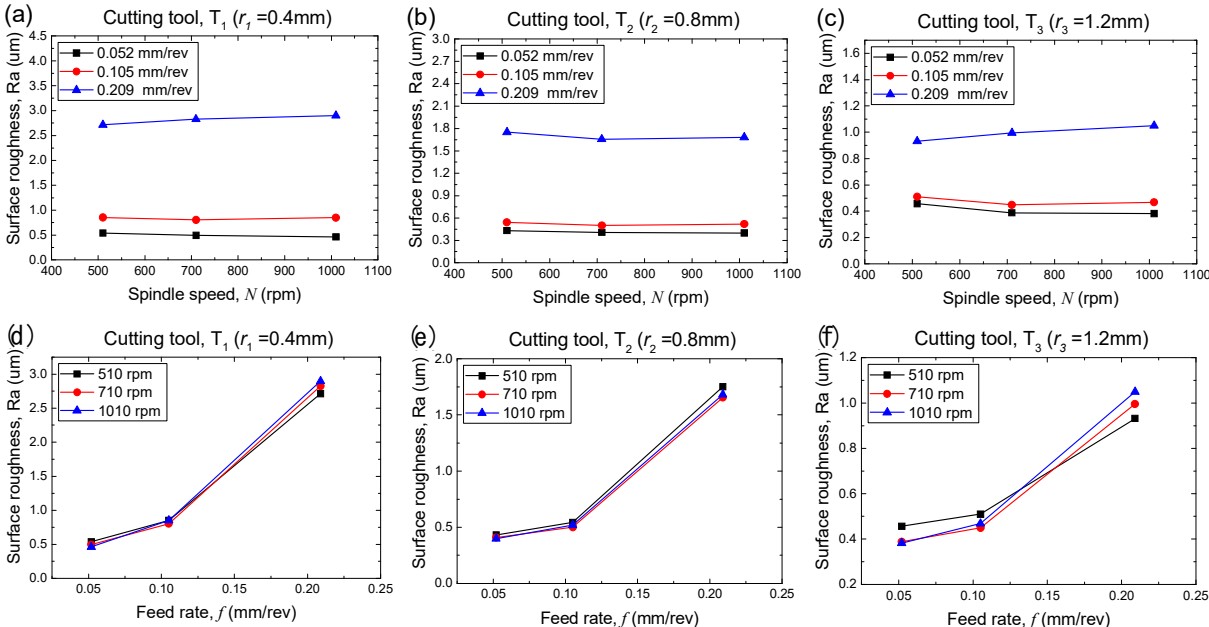

**Figure 2.** Effect of spindle speed (*N*) and feed rate (*f*) on surface roughness ($R_a$) for using different tool nose radii $r_1$, $r_2$ and $r_3$.

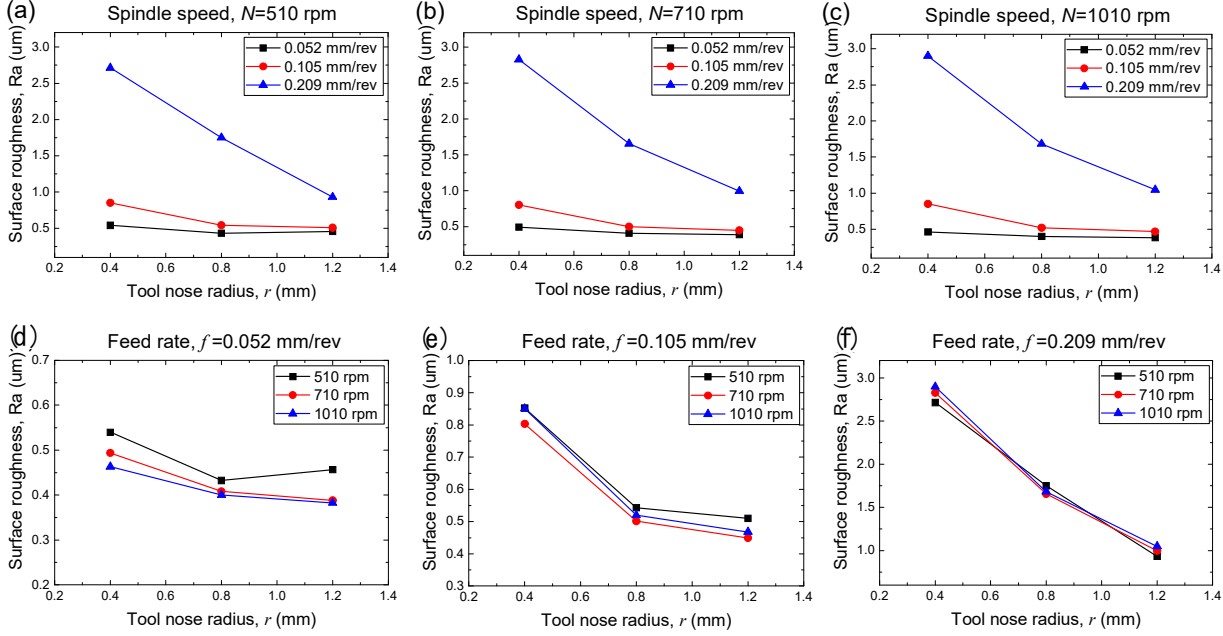

**Figure 3.** Effect of tool nose radius ($r_1$, $r_2$, $r_3$) on surface roughness ($R_a$) for different (**a–c**) spindle speeds (*N*) and (**d–f**) feed rates (*f*).

### 3.2. ANOVA, Linear Regression Model and S/N Ratio

Table 3 shows the analysis of variance (ANOVA) results for three different tool nose radii ($r_1$, $r_2$, $r_3$). As can be seen, the feed rate ($f$) is a significant effect parameter for surface roughness ($R_a$) with the utilization of the tool nose radius $r_1$, $r_2$ and $r_3$. However, the spindle speed indicates that it is an insignificant parameter for surface roughness ($R_a$) with all cutting tools. These results show good agreement with Carou's research [1], which describes that feed rate has the lowest P-Value (less than 0.05) for the dry machining test. Thus, it is a major factor for determining the machined surface quality of magnesium alloy. Dutta et al. [31] investigated the turning parameters for the dry cutting of a magnesium workpiece using the Taguchi methodology. It was concluded that an ANOVA had found that the feed rate and depth of cut has a considerable effect on the surface roughness and cutting force, while the cutting speed was implied as a normal effect parameter (P-Value = 0.43). Table 4 shows the ANOVA of surface roughness ($R_a$) for the combination of cutting parameters $N$, $f$ and $r$ (CNFR). It is used to observe the effect of the tool nose radius on surface roughness. Based on Table 4, the feed rate ($f$) and tool nose radius ($r$) were revealed as significant effect parameters for surface roughness ($R_a$), while the spindle speed ($N$) is an insignificant parameter. The linear regression model is the algebraic representation of the regression line, which explains the correlation between the cutting parameters and response (surface roughness ($R_a$)). The linear regression equation is employed to forecast the experimental measurement of surface roughness ($R_a$). As a result, the coefficient of determination ($R^2$) of the linear regression model for surface roughness ($R_a$) using different tool nose radii $r_1$, $r_2$, $r_3$ and CNFR were revealed as 99.97%, 99.96%, 99.71% and 99.08%, respectively. Figure 4a–d indicates the comparison between the values of experimental measurements and the regression model for surface roughness ($R_a$) using three different tool nose radii, $r_1$, $r_2$, $r_3$ and CNFR, respectively. The values of the experimental measurement and regression model are statistical similar, which confirms the excellent correlation between them [32]. Thus, the linear regression model is valid to forecast the surface roughness ($R_a$) for each tool nose radius, $r_1$, $r_2$, $r_3$ and CNFR, during the dry turning process of the magnesium alloy. In this study, the linear regression equations for surface roughness with different cutting tools were obtained as in the following equations:

Linear regression equations for different tool nose radius:

$r_1$

$$R_a = 0.927 - 0.000539\,N - 8.75\,f + 0.003269\,N.f + \left(1.38 \times 10^{-7}\right) N^2 + 80.68\,f^2 \tag{2}$$

$r_2$

$$R_a = 0.984 - 0.000971\,N - 6.913\,f - 0.000417\,N.f + \left(6.18 \times 10^{-7}\right) N^2 + 58.97\,f^2 \tag{3}$$

$r_3$

$$R_a = 0.883 - 0.000841\,N - 4.213\,f + 0.002388\,N.f + \left(3.61 \times 10^{-7}\right) N^2 + 23.56\,f^2 \tag{4}$$

*CNFR*

$$R_a = 0.664 - 0.00072\,N + 4.82\,f - 0.406\,r + 0.00175\,N.f - \left(8.58 \times 10^{-5}\right) N.r - 14.311\,f.r \\ + \left(3.73 \times 10^{-7}\right) N^2 + 54.4\,f^2 + 0.793\,r^2 \tag{5}$$

**Table 3.** ANOVA of surface roughness ($R_a$) for various tool nose radius.

| Tool Nose Radius | Source | DF | Adj SS | Adj MS | F-Value | *p*-Value | Remarks |
|---|---|---|---|---|---|---|---|
| | $N$ (rpm) | 2 | 0.00218 | 0.00109 | 0.22 | 0.812 | Insignificant |
| $r_1$ (0.4 mm) | $f$ (mm/rev) | 2 | 9.37852 | 4.68926 | 946.75 | 0.000 | Significant |
| | Error | 4 | 0.01981 | 0.00495 | - | - | - |
| | Total | 8 | 9.40051 | - | - | - | - |
| | $N$ (rpm) | 2 | 0.00471 | 0.00236 | 6.22 | 0.059 | Insignificant |
| $r_2$ (0.8 mm) | $f$ (mm/rev) | 2 | 3.03588 | 1.51794 | 4009.63 | 0.000 | Significant |
| | Error | 4 | 0.00151 | 0.00038 | - | - | - |
| | Total | 8 | 3.04211 | - | - | - | - |
| | $N$ (rpm) | 2 | 0.000935 | 0.000467 | 0.17 | 0.850 | Insignificant |
| $r_3$ (1.2 mm) | $f$ (mm/rev) | 2 | 0.609613 | 0.304806 | 110.62 | 0.000 | Significant |
| | Error | 4 | 0.011022 | 0.002755 | - | - | - |
| | Total | 8 | 0.621569 | - | - | - | - |

**Table 4.** ANOVA of surface roughness ($R_a$) for the combination of cutting parameters $N$, $f$, $r$ (CNFR).

| Material | Source | DF | Adj SS | Adj MS | F-Value | *p*-Value | Remarks |
|---|---|---|---|---|---|---|---|
| | $N$ (rpm) | 2 | 0.003 | 0.00148 | 0.01 | 0.989 | Insignificant |
| | $f$ (mm/rev) | 2 | 10.3923 | 5.19616 | 38.94 | 0 | Significant |
| Magnesium | $r$ (mm) | 2 | 2.6759 | 1.33796 | 10.03 | 0.001 | Significant |
| | Error | 20 | 2.6689 | 0.13345 | - | - | - |
| | Total | 26 | 15.7401 | - | - | - | - |

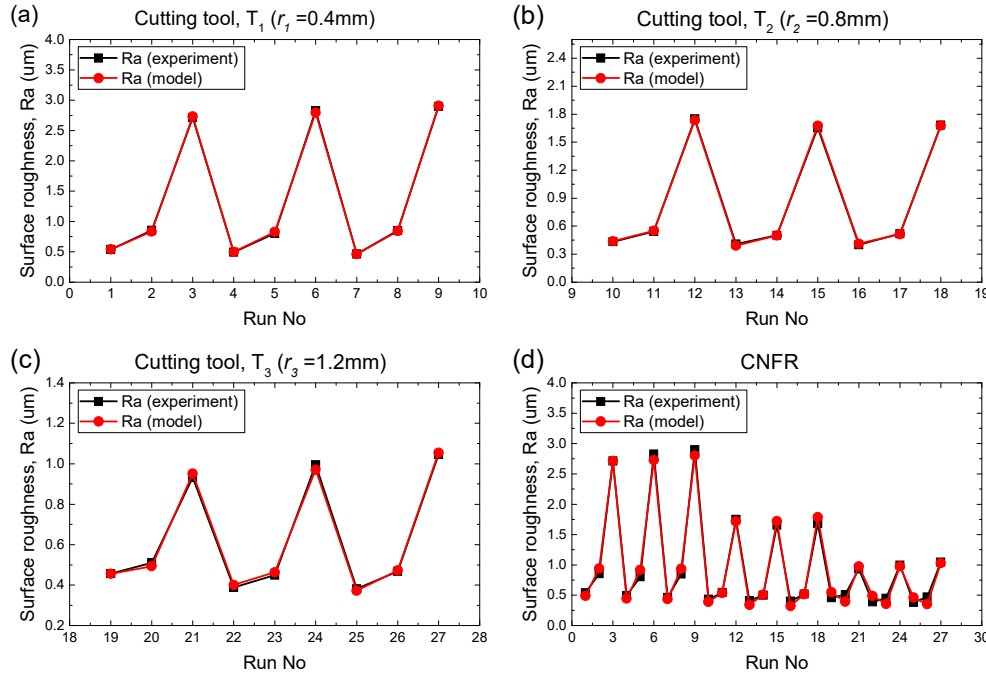

**Figure 4.** Comparison between values of experimental measurement and the prediction model for surface roughness ($R_a$) using the following three different tool nose radii: (**a**) $r_1$, (**b**) $r_2$, (**c**) $r_3$ and (**d**) CNFR.

The main effect plots for the S/N ratio illustrate the effect of cutting parameters on surface roughness ($R_a$) and optimize the experimental conditions for the turning process. According to the Taguchi methodology, the highest S/N ratio value in the levels of cutting parameters demonstrates the optimal experimental condition for minimum surface roughness ($R_a$) [33,34]. Furthermore, if there is no slope of the linear line for different levels (the

linear line reveals in a parallel direction to the *X*-axis), it means that the cutting parameters do not affect surface roughness ($R_a$). The higher slope of the linear line between cutting parameters also shows a higher effect on surface roughness [35]. Figure 5a–c shows the S/N ratio versus the cutting parameter levels (*N*, *f*) for surface roughness ($R_a$) in different tool nose radii, $r_1$, $r_2$ and $r_3$, respectively. Figure 5a–c show that the effect of feed rate (*f*) on surface roughness ($R_a$) is higher than spindle speed (*N*). The level one of feed rate (*f*) shows the optimal cutting condition for surface roughness ($R_a$). These results reveal good agreement with a study by Kolluru et al. [36] for the dry turning operation of an AM magnesium alloy. However, the level two of spindle speed (*N*) shows the best cutting condition for surface roughness. Thus, the level one of feed rate (*f*) and level two of spindle speed (*N*) were combined as an optimal experimental condition for surface roughness ($R_a$) during the dry turning process using tool nose radii of $r_1$, $r_2$ and $r_3$. The S/N ratio values revealed in Figure 5a–c are summarized in Table 5. The Delta shows the scale of effect on surface roughness, which is calculated from the highest and lowest value of each cutting parameter level. The rank is used to indicate which cutting parameters have a greater effect on the surface roughness ($R_a$). For instance, the feed rate (*f*) reveals in rank one, which explains that the feed rate (*f*) performs as the greatest effect on surface roughness ($R_a$). Figure 5d demonstrates the S/N ratio versus the cutting parameter levels (*N*, *f*, *r*) for surface roughness ($R_a$). The feed rate (rank one) performs as the greatest effect on surface roughness ($R_a$), followed by tool nose radius (rank two) and spindle speed (rank three). The highest S/N ratio value of the spindle speed (*N*), feed rate (*f*) and tool nose radius (*r*) are illustrated in level two, level one and level three, respectively. Therefore, the optimal cutting condition during the dry turning process for magnesium alloy is the following combination of parameters: level two of spindle speed (*N*), level one of feed rate (*f*) and level three of tool nose radius (*r*). The S/N ratio values shown in Figure 5d are summarized in Table 6.

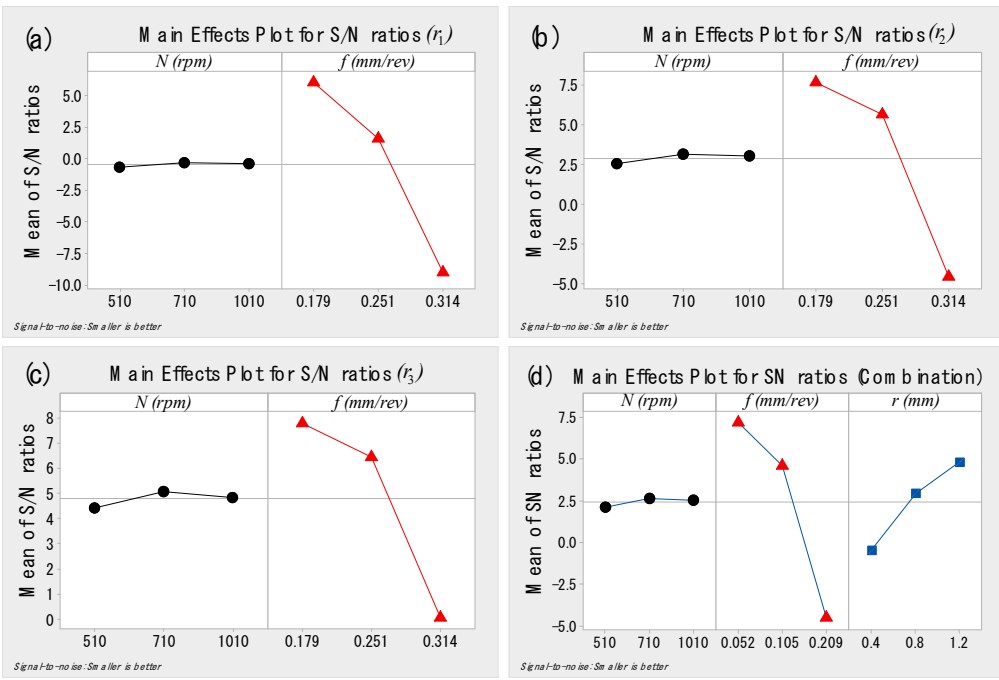

**Figure 5.** Main effect plot for average S/N ratios versus cutting parameters for surface roughness ($R_a$) in the following different tool nose radii: (**a**) $r_1$, (**b**) $r_2$, (**c**) $r_3$ and (**d**) CNFR.

**Table 5.** S/N ratio response table for surface roughness (R$_a$) with different tool nose radius (*r*).

| Tool Nose Radius (*r*) | Levels | Cutting Parameters | |
|---|---|---|---|
| | | Spindle Speed (*N*) | Feed Rate (*f*) |
| | 1 | −0.6437 | 6.0553 |
| | 2 | −0.3323 | 1.5649 |
| $r_1$ | 3 | −0.3839 | −8.9802 |
| | Delta | 0.3114 | 15.0355 |
| | Rank | 2 | 1 |
| | 1 | 2.575 | 7.674 |
| | 2 | 3.133 | 5.660 |
| $r_2$ | 3 | 3.040 | −4.586 |
| | Delta | 0.558 | 12.260 |
| | Rank | 2 | 1 |
| | 1 | 4.42276 | 7.79418 |
| | 2 | 5.06990 | 6.46410 |
| $r_3$ | 3 | 4.85002 | 0.08441 |
| | Delta | 0.64714 | 7.70977 |
| | Rank | 2 | 1 |

**Table 6.** S/N ratio response table for surface roughness (R$_a$) with the CNFR.

| Levels | Cutting Parameters | | |
|---|---|---|---|
| | Spindle Speed (*N*) | Feed Rate (*f*) | Tool Nose Radius (*r*) |
| 1 | 2.1180 | 7.1745 | −0.4533 |
| 2 | 2.6236 | 4.5632 | 2.9161 |
| 3 | 2.5021 | −4.4940 | 4.7809 |
| Delta | 0.5055 | 11.6685 | 5.2342 |
| Rank | 3 | 1 | 2 |

### 3.3. Micro-Image of Machined Surface Topography

Figure 6a–c shows the SEM micro-images of the machined surface affected by the different spindle speeds of 510, 710 and 1010 rpm using a tool nose radius $r_1$ at the lowest feed rate of 0.052 mm/rev, respectively. The SEM machined surface topographies were investigated in different scales at 100× and 1000× magnification. From the scale 100× magnification, the machined surface topographies do not show feed marks clearly. At spindle speeds of 510 and 710 rpm (Figure 6a,b), the micro-voids appeared on the machined surface, while the favorable surface roughness was generated at the spindle speed of 1010 rpm (Figure 6c). Fernández-Abia et al. [37] also reported that cavities and metal debris with smeared material particles were revealed on the machined surface at low cutting speeds from 37 to 300 rpm. At high cutting speeds from 450 to 870 rpm, cavities and metal debris were not present on the machined surface. From the scale 1000× magnification, the material debris and material side were observed. However, the machined surface topographies do not appear to be much different according to the change of spindle speed (*N*). Figure 7a–c illustrates the SEM micro-image of the machined surface that is affected by different feed rates of 0.052, 0.105 and 0.209 mm/rev, respectively, using a tool nose radius $r_1$ at a spindle speed of 710 rpm. From the scale 100× magnification, the feed marks gradually and clearly appear with an increase in the feed rate (*f*). Moreover, the feed mark ridges are revealed in the longer distance with an increase in the feed rate. From the scale 1000× magnification, at the feed rate of 0.052 mm/rev (Figure 7a), the small material sides and material debris are demonstrated on the machined surface topography, and they are larger according to the increase in the feed rate (Figure 7b,c). As mentioned above [29], the increase in the feed rate caused the increase in friction, then led to the production of a rough surface. The surface topographies illustrate a significant difference with the change of feed rate (*f*), as shown in Figure 7. The surface quality was improved at the lower feed

rate (*f*). Figure 8a–c shows the micro-image of the machined surface using tool nose radii of $r_1$, $r_2$ and $r_3$, respectively, with the combination of cutting parameters of 710 rpm and 0.209 mm/rev. From the scale 100× magnification, as shown in Figure 8a, the feed marks are clearly revealed on the machined surface topography, and gradually disappear with a larger tool nose radius (Figure 8b,c). From the scale 100× magnification, Figure 8a shows the large material side and material debris on the machined surface topography during the turning process using the tool nose radius $r_1$. The material side and material debris become smaller and less when the turning process was performed with a larger tool nose radius (Figure 8b,c). As shown in Figure 8, the machined surface was significantly improved by using a larger tool nose radius (*r*). These results agree with those of Dutta et al. [16]. These researchers described that the reduction in the tool nose radius results in defects and a rise in roughness on the turned surface (SEM image) due to the decrease in the tooltip strength and the lower temperature in the cutting zone. This leads to plowing grooves, smears, cracks and metal debris.

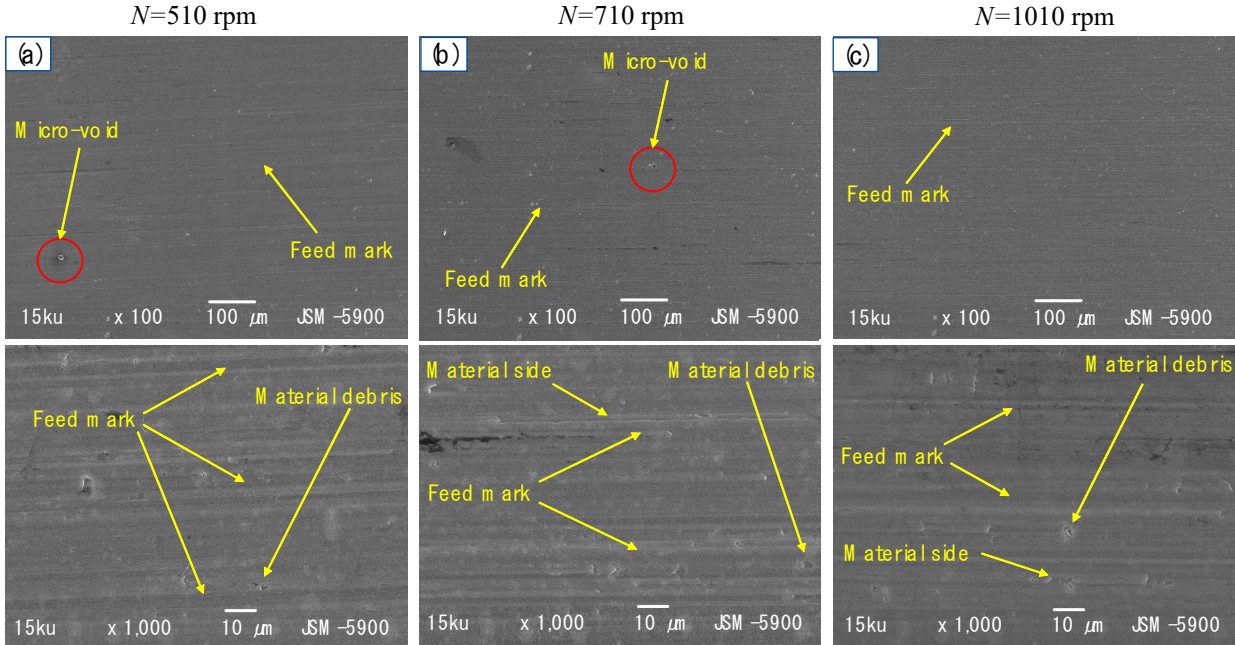

**Figure 6.** Micro-image of machined surface for spindle speeds of (**a**) 510, (**b**) 710 and (**c**) 1010 rpm at 100× and 1000× magnifications (SEM) with a combination of cutting parameters $r_1$ and 0.052 mm/rev.

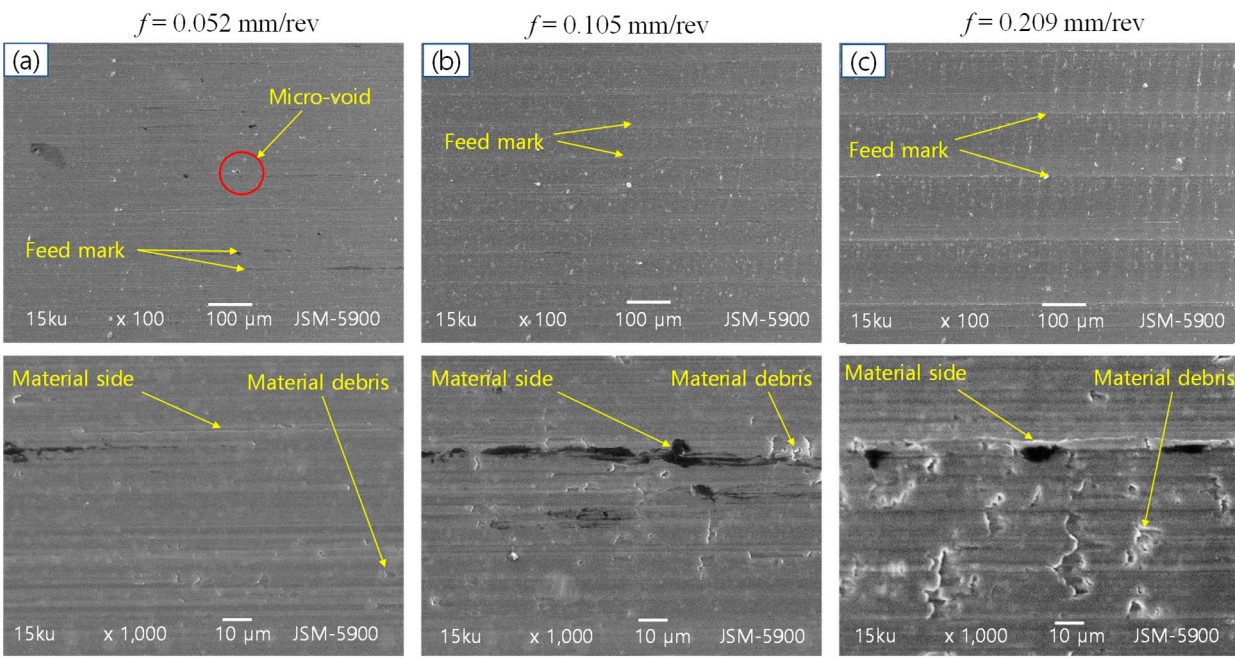

**Figure 7.** Micro-image of machined surface for feed rates of (**a**) 0.052, (**b**) 0.105 and (**c**) 0.209 mm/rev at 100× and 1000× magnifications (SEM) with a combination of cutting parameters $r_1$ and 710 rpm.

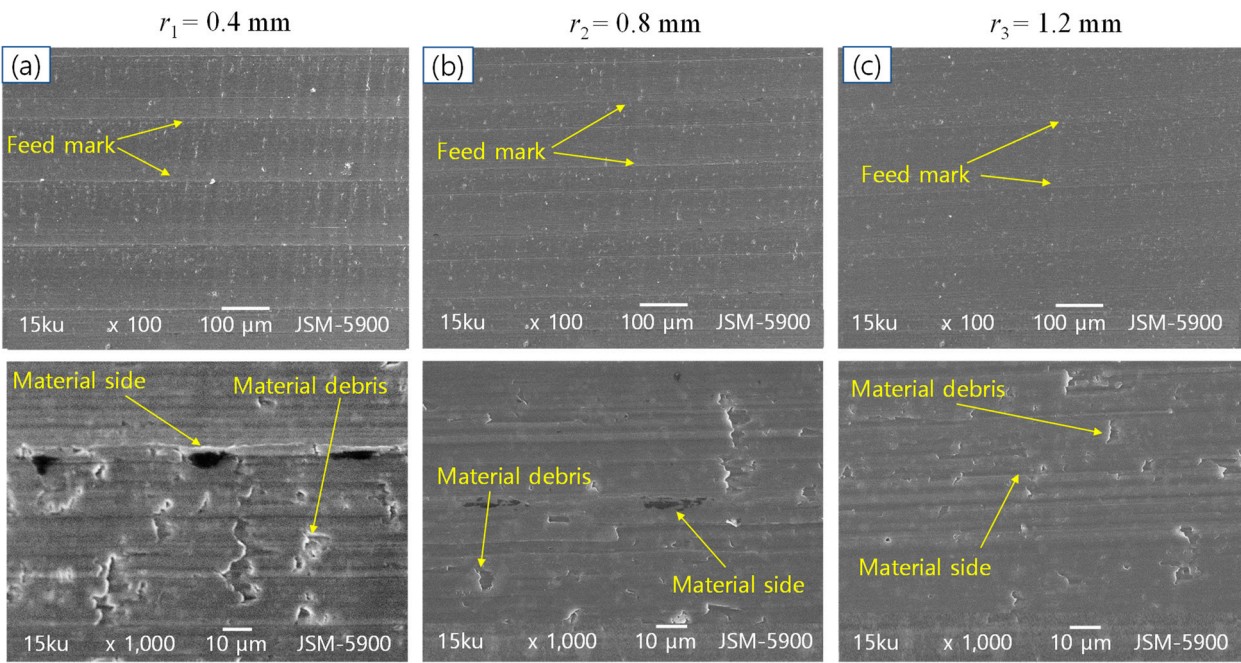

**Figure 8.** Micro-image of machined surface for tool nose radii of (**a**) $r_1$ = 0.4, (**b**) $r_2$ = 0.8 and (**c**) $r_3$ = 1.2 mm at 100× and 1000× magnifications (SEM) with a combination of cutting parameters 710 rpm and 0.209 mm/rev.

## 4. Conclusions

In this research, the dry turning process is carried out on the minor scale diameter of biodegradable magnesium alloys using a different tool nose radius. The statistical analysis and micro-images of the machined surface topography were used to investigate and analyze the cutting parameters to improve the machined surface quality. The conclusions of the research were summarized as follows:

1.  The change of feed rate and tool nose radius revealed a strong change of surface roughness. However, surface roughness slightly changed with the change of spindle

speed. Thus, the feed rate and tool nose radius are revealed as cutting parameters that have a strong effect on surface roughness.

2. The ANOVA demonstrated that the feed rate has a significant effect on surface roughness for each tool nose radius, $r_1$, $r_2$ and $r_3$. From the analysis of the CNFR, the feed rate and tool nose radius performed as a significant effect parameter for surface roughness. According to the results of the regression model, linear regression models can be acceptably employed to forecast the surface roughness for each tool nose radius and CNFR during the dry turning process of magnesium alloy. The S/N ratio indicated that the cutting parameters for optimum surface roughness are a combination of a spindle speed of 710 rpm and a feed rate of 0.052 mm/rev for each tool nose radius. In the case of analysis with the CNFR, the optimum surface roughness is achieved with a combination of a spindle speed of 710 rpm, a feed rate of 0.052 mm/rev and a tool nose radius of 1.2 mm.

3. According to the SEM micro-image, the machined surface topography was not strongly affected by the change of spindle speed when compared with feed rate and tool nose radius. The feed marks, material side and material debris become larger in size and higher in number according to the increase in the feed rate and a smaller tool nose radius. The better machined surface topography appeared with machining at a lower feed rate, larger tool nose radius and higher spindle speed.

4. This study has good results that show potential use for academic research and industrial applications. Moreover, it is very helpful in implant, orthopedic and trauma surgery applications to ensure a high-quality surface as well as long service life in the human body.

**Author Contributions:** Conceptualization, H.-G.S. and S.H.; methodology, S.H.; software, H.-C.J. and W.-H.S.; validation, H.-G.S., H.-C.J. and W.-H.S.; formal analysis, S.H.; investigation, H.-G.S. and S.H.; resources, S.H.; data curation, S.H.; writing—original draft preparation, S.H.; writing— review and editing, H.-G.S. and S.H.; visualization, H.-C.J. and W.-H.S.; supervision, H.-G.S.; project administration, H.-C.J., W.-H.S. and H.-G.S.; funding acquisition, H.-G.S. and S.H. All authors have read and agreed to the published version of the manuscript.

**Funding:** This research was supported by Basic Science Research Program through the National Research Foundation of Korea (NRF) funded by the Ministry of Education (2020R1I1A1A01069429).

**Conflicts of Interest:** The authors declare no conflict of interest.

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
