# Peer review of "Investigation of Surface Quality for Minor Scale Diameter of Biodegradable Magnesium Alloys during the Turning Process Using a Different Tool Nose Radius"

_metals, doi:10.3390/met11081174_

Round 1
Reviewer 1 Report
This is a well-written manuscript on an interesting topic which I am happy to recommend for publication in Metals.
One minor point to be addressed: some of the letters/numbers/graphs in Figures 1, 6-8 are difficult to read.
Author Response
Dear Editors and Reviewer,
First, we would like to thank the editors and reviewers for your careful reading of my submitted manuscript. According to your comments, the yellow highlights were appeared on the modified or added statements in the manuscript. The review report and manuscript revisions were uploaded with a PDF file.
Best regards,
Hyung-Gon Shin

Reviewer 2 Report
In this paper, the dry turning process is carried out on the minor scale diameter of biodegradable magnesium alloys using various tool nose radius, feed rates, spindle speed. The ANOVA, S/N ratio, linear regression model and micro-images of the machined surface topography were used to investigate and analyze the cutting parameters to improve the machined surface quality, which is a nice discussion. Moreover, the cutting parameters for optimum surface quality were founded, and the prediction mathematical models of surface roughness were also obtained. This work not only provides valuable guidelines for the dry turning process of biodegradable magnesium alloy micro-cylinders, but also boosts the application of magnesium alloy in implant, orthopaedic and trauma surgery fields.
However, some points also need to revise as follows:
- It is advised to provide the heat treatment status, hardness, and optical microstructure of magnesium alloy samples.
- Please check the typed errors like ‘0.105 rpm’ in ‘Figure 6. Micro-image of machined surface for spindle speed (a) 510 rpm, (b) 0.105 rpm and (c) 1010 rpm at ×100 and ×1000 magnifications (SEM) with combination of cutting parameter r1 and 0.052 mm/rev.’
- Please check conclusion 5 ‘The feed marks, material side and material debris become larger in size and increase in number according to the increase of feed rate and larger tool nose radius’.
Author Response
Dear Editors and Reviewer,
First, we would like to thank the editors and reviewers for your careful reading of my submitted manuscript. According to your comments, the yellow highlights were appeared on the modified or added statements in the manuscript. The review report and manuscript revisions were uploaded with PDF file and word file, respectively.

Reviewer 3 Report
The authors conducted a deep Investigation on the surface quality of bio-degradable magnesium alloys. The results are of interest and significance. The manuscript can be accepted after a minor revision.
The conclusion is too long to focus on the major contribution or interesting findings. It should be shortened.
Author Response
Dear Editors and Reviewer,
First, we would like to thank the editors and reviewers for your careful reading of my submitted manuscript. According to your comments, the yellow highlights were appeared on the modified or added statements in the manuscript. The review report and manuscript revisions were uploaded with PDF file and word file, respectively.
Best regards,
Hyung-Gon Shin

Reviewer 4 Report
The study deals with interesting and important issue of magnesium alloy machining. in this study the influence of performance parameters on the surface roughness was investigated.
The paper was prepared rather well, but revision is needed. Below are my detailed comments:
- In the materials and methods section there is a problem with subsection numbering (the subsection 2.1 appears two times)
- in the materials and methods section the new subsection should be added: section 2.3. Statistical analysis, in which the methods used in the statitical analysis will be described. You can put in this section some text from the lines 190-198
- the regression equation of Ra for the combination of cutting parameters N, f, r should be presented in the manuscript,
Author Response

(The authors gave the same response as above.)

Round 2
Reviewer 2 Report
The manuscript is improved.
Reviewer 4 Report
Dear Authors,
thank you very much for provided changes. The manuscript was significantly improved and all my comments were adressed.
Congratulations!